# Effects of Preventive Administration of Propylene Glycol or Sucrose in Dairy Cows with Elevated Blood Non-Esterified Fatty Acids During the Close-Up Period

**DOI:** 10.3390/ani15213211

**Published:** 2025-11-04

**Authors:** Kyoko Chisato, Miki Ishizaka, Takumi Honjo, Yuta Watanabe, Rika Fukumori, Shin Oikawa

**Affiliations:** Veterinary Herd Health, Department of Veterinary Medicine, School of Veterinary Medicine, Rakuno Gakuen University, Ebetsu 069-8501, Hokkaido, Japan; k-chisato@rakuno.ac.jp (K.C.); fukumori@rakuno.ac.jp (R.F.)

**Keywords:** dairy cow, close-up period, propylene glycol, sucrose, prophylactic administration

## Abstract

**Simple Summary:**

The objective of this study was to evaluate the effects of administration of propylene glycol (PG) or sucrose (SC) on health and production outcomes in dairy cows with elevated non-esterified fatty acids (NEFA) levels of 0.3 mEq/L or higher during the close-up period. Thirty-five cows from two farms in Hokkaido were assigned to PG, SC, or untreated control groups, with treatments administered for 5 days starting from the blood testing. In PG and SC cows, blood profiles related to energy metabolism, including NEFA and β-hydroxybutyrate concentrations, improved after calving compared with controls, and liver function was maintained as well. Cows in both treatment groups exhibited significant decreases in postpartum culling rates. These findings suggest that prophylactic administration of PG or SC may contribute to postpartum productivity.

**Abstract:**

The purpose of this study was to evaluate the preventive effects of propylene glycol (PG) or sucrose (SC) in dairy cows with high levels of non-esterified fatty acids (NEFAs) during the close-up period. From July 2021 to August 2022, blood samples were collected from 193 cows between 14 and 7 days prior to the expected calving date in two farms, and 35 multiparous cows with serum NEFA ≥ 0.3 mEq/L were randomly assigned to PG (500 mL/day, *n* = 11), SC (1000 mL/day of 50% solution, *n* = 11), and untreated control (HC; *n* = 13) groups. Treatments were administered orally for 5 consecutive days. Compared with HC cows, the serum NEFA concentration tended to be lower in SC cows at 3 days in milk (DIM) and was significantly lower in PG cows at 14 DIM. Serum β-hydroxybutyrate concentrations tended to be lower in SC cows at 21 DIM. Blood glucose concentrations were higher in both treatment groups at 3 DIM, and the serum total bilirubin concentration remained lower until 14 DIM in PG cows and until 7 DIM in SC cows. At 7 DIM, PG cows showed significantly higher total very low-density lipoprotein levels and PG and SC cows had significantly or tendentially higher low-density lipoprotein triglyceride concentrations. Cows in both treatment groups had significantly reduced culling after calving. These results suggest that prophylactic administration of PG or SC improves energy metabolism by supporting liver function, thereby reducing postpartum culling, with the PG group showing a more consistent effect.

## 1. Introduction

The three weeks before and after calving are called the “transition period” [1,2], during which dramatic changes in energy occur in dairy cows. This means that cows have an increased nutritional demand for fetal growth and the production of colostrum and milk. During this period, cows readily fall into a negative energy balance (NEB) when the energy supply from feed intake is insufficient or when conditions exist that disturb feed intake [3,4]. The cow attempts to adapt to NEB by utilizing the carbohydrates, proteins, and lipids stored in the body. However, if these metabolic adaptations are not sufficiently successful, peripartum diseases such as ketosis, retained placenta, and displaced abomasum may occur. The occurrence of peripartum diseases not only leads to reproductive disorders, such as decreased conception rates [4], but also increases the risk of early culling, death, and sale, resulting in significant economic losses for dairy farms [5,6].

Serum non-esterified fatty acids (NEFA) and β-hydroxybutyrate (BHBA) concentrations are known to be markers of the degree of NEB [7,8]. Measuring NEFA concentrations before calving has been reported to be useful for predicting postpartum diseases and culling [9,10,11]. On the other hand, excessive influx of NEFA into the liver generates reactive oxygen species (ROS) during energy metabolism, thereby exacerbating oxidative stress [12]. Consequently, the risk of peripartum diseases such as fatty liver and ketosis is markedly increased, and these conditions have also been reported to be closely associated with reduced productivity [13,14]. Therefore, identifying cows with elevated NEFA concentrations before calving and implementing early prophylaxis would be beneficial for dairy herd management. However, effective preventive strategies for prepartum NEB cows have not yet been established in the field.

Propylene glycol (PG) has long been used to treat NEB in dairy cows. The gluconeogenic effect of PG in ruminants has been recognized since the 1950s [15]. PG is metabolized to propionate in the rumen, which is then converted into glucose as a gluconeogenic substrate via the liver [16]. These actions lead to a decrease in the blood concentrations of NEFA and BHBA [17].

In our laboratory, we have focused on sucrose (SC) in addition to PG. In Japan, PG requires a veterinary prescription, whereas SC can be purchased without that procedure and is more readily available. SC is known to be hydrolyzed into monosaccharides by rumen microbes, leading to the production of volatile fatty acids (VFA) such as propionate and butyrate. Similarly to PG, SC is expected to be effective in reducing NEB by promoting gluconeogenesis and stimulating insulin secretion. Although previous studies have investigated the effects of including SC in the diet on milk production and rumen fermentation [18,19], its impact on blood lipid and carbohydrate metabolism remains unclear when orally administered prophylactically to cows with NEB, as with PG.

To date, studies have investigated the effects of prepartum administration of both substances on blood biochemical parameters and productivity [19,20], as well as the influence of PG on hepatic lipidosis [21,22]. In addition, experimental studies have examined how PG administration influences energy metabolism in cows subjected to 50% feed restriction [23,24]. However, the first two of the above studies were conducted in dry cows with normal NEFA concentrations, whereas the third study evaluated cows under experimentally induced low-energy conditions. Although these studies provide some guidance for preventive strategies, directly applying their findings to high-NEFA cows encountered in field conditions remains challenging. Moreover, in field study, identifying and monitoring high NEFA cows is a complex and time-consuming endeavor, which contributes to the limited availability of research data.

Therefore, this study was conducted with a primary focus on practical observation. It aimed to clarify how prophylactic administration of PG or SC for a fixed duration during the close-up period to dairy cows exhibiting high NEFA concentrations influences periparturient blood profiles, oxidative stress status, the incidence of postpartum disease, culling rate, and milk production.

## 2. Materials and Methods

### 2.1. Animals

This study was conducted between July 2021 and August 2022 at 2 commercial dairy farms in Kitahiroshima and Ebetsu, Hokkaido, Japan, each housing approximately 200 dairy cows. All procedures used in this study were approved by the Animal Experiment Committee of Rakuno Gakuen University (approval #VH21C7). The management and feeding systems at each farm were as follows. One farm in Kitahiroshima (Farm A) kept both dry and lactating cows housed in free stalls (FS). Dry cows were fed a total mixed ration (TMR) once daily at 09:00, and lactating cows at 10:30. The TMR for lactating cows was formulated to contain 15.6% crude protein (CP), 22.1% neutral detergent fiber (NDF), and 1.77 Mcal/kg of net energy for lactation (NEl), whereas the TMR for dry cows contained 12.0% CP, 40.3% NDF, and 1.39 Mcal/kg NEl. Another farm in Ebetsu (Farm B) included dry cows housed in free barns (FB), while lactating cows were kept in FS. Lactating cows were fed a partial mixed ration (PMR; 17.1% CP, 37.9% NDF, 1.61 Mcal/kg NEl) once daily at 09:00. Additionally, concentrate (20.6% CP, 14.0% NDF, 1.67–1.71 Mcal/kg NEl) was provided in the automatic milking system at 2 to 6 kg per day, depending on days in milk (DIM) and milk yield. Dry cows (from 21 days before the expected calving date) were fed by component feeding (15.6% CP, 46.0% NDF, 1.50 Mcal/kg NEl). Feeding programs at both farms were formulated to meet the energy requirements of dairy cows as recommended by the National Research Council [25].

The study enrolled 193 multiparous Holstein cows (Farm A: *n* = 110; Farm B: *n* = 83), that were prospectively monitored from 14 to 7 days before the expected calving date through 21 DIM. Farm records provided general animal information, including individual identification numbers, parity, expected calving dates, and actual calving dates.

### 2.2. Blood Sampling and Physical Examinations

Blood samples were collected at five points: −13 ± 1 days (mean ± standard error of mean (SEM)) based on the expected calving date (−9 ± 1 days based on the actual calving date), and on days 3 (3 ± 1 days), 7 (7 ± 1 days), 14 (14 ± 1 days) and 21 (21 ± 1 days) postpartum. Sampling was performed via the coccygeal vein at 10:40 at Farm A and at 14:00 at Farm B. Blood samples were collected into plain collection tubes (Venoject II VP-AS109K, Terumo Corporation, Tokyo, Japan) and immediately stored at 4 °C. For serum separation, centrifugation was performed at 2000× *g* for 15 min within 1 h of collection. Serum for the isolation of lipoprotein was stored at 4 °C, whereas all other samples were stored at −20 °C until analysis.

The body condition score (BCS) [26] and rumen fill score (RFS) [27] were determined at the time of blood collection. The BCS was rated on a 5-point scale with increments of 0.25, where higher scores indicated greater body fat stores. Furthermore, the RFS was assessed by observing the left hind flank of each cow from the rear and evaluating the skin line of the paralumbar fossa and the inferior transverse process (lumbar vertebral processes) to determine tension due to dry matter intake (DMI). This parameter was rated on a scale from 1 to 5 (1 = very poor, 2 = poor, 3 = good, 4 = very good, 5 = excellent).

### 2.3. Cows Eligible for Preventive Treatment

Cows without clinical abnormalities and with blood NEFA concentrations ≥ 0.3 mEq/L between 14 and 7 days prior to the expected calving date (*n* = 52) were classified as high-NEFA cows, reflecting an elevated risk of peripartum diseases [11]. The high-NEFA cows were randomly assigned to three groups: propylene glycol administration (PG), sucrose administration (SC), and untreated controls (HC). In the PG cows, 500 mL of propylene glycol (Neornogen, Kyoritsu Seiyaku Corporation, Tokyo, Japan) was orally administered once daily. For the SC cows, 1000 mL of 50% SC solution (sucrose, Nippon Beet Sugar Mfg. Co., Ltd., Tokyo, Japan) was administered once daily. A 50% SC solution was prepared by dissolving 500 g of sucrose in 500 mL of water. Both treatments commenced on the day of initial blood sampling (preventive day 1) and planned to be continued for 5 consecutive days. The trial was discontinued at the calving day for cows that delivered before completing the regimen (mean treatment duration ± SE: 4.8 ± 0.1 days).

Of the 52 high-NEFA cows, 17 that delivered twins were excluded from the study due to the reported association between twin pregnancies and increased prepartum NEFA concentrations [28]. Consequently, 35 cows were selected for the final analysis: 13 HC cows, 11 PG cows and 11 SC cows.

In contrast, cows with blood NEFA concentrations of <0.3 mEq/L between 14 and 7 days prior to the expected calving date (*n* = 141) were classified as healthy and excluded from this preventive trial. Of these, 15 cows matched to the preventive treatment cows by sampling date and parity were selected as reference negative controls (NC).

### 2.4. Separation of Lipoproteins

Sera were used immediately for lipoprotein separation without freezing. Lipoprotein separation was performed based on the method described previously [29], using a fixed-angle rotor (TLA-X110, Beckman Coulter, Fullerton, CA, USA) in an ultracentrifuge (Optima^TM^TLX Ultracentrifuge, Beckman Coulter) at 16 °C. Very low-density lipoprotein (VLDL; 657,000× *g* for 2 h 55 min, d < 1.006), low-density lipoprotein (LDL; 657,000× *g* for 4 h 20 min, d < 1.063), and high-density lipoprotein (HDL; 657,000× *g* for 7 h 15 min, d < 1.210) were collected. The separated lipoprotein fractions were stored at −20 °C until measurement of their principal component concentrations.

### 2.5. Measurement of Serum Biochemical Parameters and Components of Lipoproteins

The assessments of serum biochemical parameters and VLDL, LDL, and HDL components were performed at I.B Co., Ltd., Osaka Saito Laboratory (Ibaraki, Japan) using a multi-functional automated analyzer (BIOLIS 30i; Tokyo Boeki Medisys Inc., Tokyo, Japan). The following reagents were used for the analytes: NEFA-HA Test Wako II (Fujifilm Wako Pure Chemicals Corporation, Osaka, Japan) for NEFA; Serotec 3-HB (Serotec, Sapporo, Japan) for BHBA; L-Type Wako Glu2 (Fujifilm Wako Pure Chemical Corp.) for glucose; L-Type Wako AST-J2 (Fujifilm Wako Pure Chemical Corp.) for aspartate transferase (AST); and Total Bilirubin E-HA Test Wako (Fujifilm Wako Pure Chemical Corp.) for T-Bil. The concentrations of phospholipids (PL), total cholesterol (T-Cho), and triglycerides (TG) in the lipoprotein fractions were measured using commercial kits (Fujifilm Wako Pure Chemical Corp.). The total protein concentration in each lipoprotein fraction was determined using the Lowry method [30]. Serum insulin concentrations were measured at the Sapporo Clinical Laboratory Co. (Kushiro, Hokkaido, Japan) using Lumipulse Presto Insulin (Fujirebio Inc., Tokyo, Japan), based on the chemiluminescent enzyme immunoassay (CLEIA) method and analyzed with a Lumipulse L2400 (Fujirebio Inc.). In addition, insulin resistance was assessed using the revised quantitative insulin sensitivity check index (RQUICKI), calculated according to the following formula [31].RQUICKI = 1/[log_10_(glucose (mg/dL)) + log_10_(insulin (μU/mL)) + log_10_(NEFA (mEq/L))]

### 2.6. Measurement of Serum Malondialdehyde (MDA) Concentration and Paraoxonase-1 (PON-1) Activity

MDA was measured using a commercial kit (MDA Assay Kit, Dojindo Laboratories, Kumamoto, Japan), as an indicator of serum lipid peroxides. PON-1 activity was measured as an indicator of serum antioxidant enzymes using a commercial kit (PON-1, Japan Institute for the Control of Aging, Nikken Seil Co., Ltd., Shizuoka, Japan). The ratio of PON-1 activity to the HDL-T-Cho concentration (PON-1/HDL-T-Cho), representing HDL-standardized PON-1 activity, was calculated [32]. In calculating PON-1/HDL-T-Cho, 0.02586 was multiplied by HDL-T-Cho as an adjustment factor [33].

### 2.7. Milk Production

Milk production was assessed using data from Dairy Herd Improvement (DHI) testing (Hokkaido Dairy Milk Recording & Testing Association, Sapporo, Japan). Parameters measured on the first DHI test day postpartum included milk yield, milk fat percentage, milk protein percentage, de novo fatty acids, preformed fatty acids, milk urea nitrogen (MUN), milk BHBA, and predicted 305-day milk yield. The 4% fat corrected milk (FCM) yield was calculated using the following formula [25]:FCM = (15 × milk fat percentage/100 + 0.4) × milk yield

### 2.8. Postpartum Disease Incidence and Culling Rate

The postpartum disease name and the dates for diseases occurring within 60 DIM were obtained from the database of the Livestock Mutual Aid System of the Agricultural Mutual Aid Association. This accurately reflects the traceability information of cows as required by domestic law. Additionally, information on culling due to cows having been sold, slaughtered or died (culling) within 60 DIM was collected.

### 2.9. Statistical Analysis

The concentrations of blood metabolites, insulin and components of lipoprotein, and RQUICKI, BCS and RFS were expressed at least squares means (LSM) ± SEM. A log_10_ transformation was applied to NEFA, BHBA, insulin and various components of lipoprotein. Other outcomes did not require transformation.

Statistical analyses were performed using JMP (version 13.0, SAS Institute Inc., Cary, NC, USA). The analysis model included treatment, sampling day, and interactions between the treatment and sampling day as fixed effects, with individual cows and farms treated as random effects. When a significant treatment effect was detected, multiple comparisons among groups were performed using Tukey’s test. Additionally, differences in variables at each sampling point were compared with those of the HC cows using Dunnett’s test.

Milk production data are presented as mean ± SEM. Statistical analyses were conducted using SPSS version 29.0 (IBM Japan, Ltd., Tokyo, Japan). Normality of data was assessed using the Shapiro–Wilk test. Differences among groups were evaluated using a linear mixed model. The statistical model was the following:Ypq = µ + Typep + Herdq + Cowr + epqr
where Ypq is the observed milk production data; µ is the overall mean; Typep is the fixed effect of the pth class of type (p = HC, PG, SC); Herdq is the random effect of the qth herd (q = FarmA, FarmB), Cowr is the random effect of the rth cow (r = 1–35); and epqr is the residual error. Bonferroni analysis was used when comparing groups. The chi-square test was used to compare the disease incidence and culling rate. When significance was calculated to be present by the chi-square test, adjusted residuals below −1.96 were rated as significantly low frequency of occurrence and those above +1.96 as significantly high frequency of occurrence [34]. Statistically significant differences were assessed at *p* < 0.05, and a trend was assessed at *p* < 0.10.

## 3. Results

There were no significant differences among the cows in terms of parity, days in milk at initial blood sampling or duration of treatment (Table 1).

Figure 1A–I present the longitudinal changes in physical scores and blood biochemical parameters throughout the study period. No significant differences in BCS were observed in the treated cows. However, the RFS of the SC cows tended to be lower than that of the HC cows at −14 DIM and tended to be higher at 14 DIM (*p* < 0.10).

The following parameters related to energy metabolism were evaluated: serum concentrations of NEFA and BHBA, glucose and insulin, and RQUICKI (Figure 1C–G). A significant interaction between day and treatment group was observed for NEFA concentrations. The HC cows tended to have higher NEFA concentrations than the SC cows at 3 DIM (*p* < 0.10) and had significantly higher ones than the PG cows at 14 DIM (*p* < 0.05). The BHBA concentration in the SC cows tended to be lower than in the HC cows at 21 DIM (*p* < 0.10). Serum glucose concentrations were significantly lower in the HC cows than in the SC cows at 3 DIM (*p* < 0.05) and tended to be lower than those in the PG cows (*p* < 0.10). Serum insulin concentrations in the HC cows tended to be lower than those in the SC and PG cows at 3 DIM and 7 DIM, respectively (*p* < 0.10). The RQUICKI of the HC cows tended to be higher than that of the PG cows at 3 DIM (*p* < 0.10).

Liver function was assessed using the AST activity and T-Bil concentration (Figure 1H,I). The AST activity of the SC cows tended to be lower than that of the HC cows at 7 DIM (*p* < 0.10). T-Bil concentrations in the HC cows were significantly higher than those in the PG and SC cows at 3 DIM (*p* < 0.05), tended to be higher at 7 DIM, and showed a trend similar to the PG cows at 14 DIM (*p* < 0.10).

The concentrations of the lipoprotein components and total lipoprotein concentrations before and after calving are shown for VLDL, LDL and HDL in Figure 2A–E, Figure 2F–J, and Figure 2K–O, respectively. Both VLDL-protein (Figure 2D) and VLDL-total concentration (Figure 2E) showed a significant interaction between day and treatment group, and the HC cows had significantly lower concentration than the PG cows at 7 DIM (*p* < 0.05). LDL-TG concentrations (Figure 2F) at 7 DIM were significantly higher in the PG cows and tended to be higher in the SC cows than in the HC cows (*p* < 0.05 and *p* < 0.10, respectively). The HDL-TG concentration (Figure 2K) at 3 DIM was significantly lower in the HC cows than in the PG cows (*p* < 0.05). The HDL-T-Cho concentration (Figure 2L) in the HC cows was significantly lower than that in the PG cows at −14 DIM (*p* < 0.05) and tended to be lower at 3 DIM (*p* < 0.10). Throughout the entire study period, the HDL-T-Cho concentration in the PG cows tended to be higher than in the HC cows (*p* = 0.086). At −14 DIM, HDL-PL (Figure 2M) and HDL-protein (Figure 2N) concentrations were significantly higher in the PG cows than in the HC cows (*p* < 0.05). Throughout the entire study period, the HDL-PL concentration in the HC cows tended to be lower than that in the PG cows (*p* = 0.098).

Figure 3A–C show the changes in serum MDA concentration, PON-1 activity, and the ratio of the PON-1 activity to the HDL-T-Cho concentration before and after calving. At 7 DIM, PON-1 activity was significantly lower in the SC cows than in the HC cows (*p* < 0.05). No significant differences among the treated cows were observed for the serum MDA concentration or in the ratio of the PON-1 activity to the HDL-T-Cho concentration.

Table 2 presents a comparison of milk yield and milk composition at the first postpartum DHI test. MUN levels were significantly higher in the SC cows than in the HC cows (*p* = 0.012). Additionally, the predicted 305-day milk yield tended to be higher in the PG cows than in the SC cows (*p* = 0.067). No significant differences were observed among the treated cows in terms of milk yield, milk fat percentage, milk protein percentage, de novo fatty acids, preformed fatty acids, the milk BHBA concentration or 4% FCM.

Table 3 presents the culling rates and incidences of postpartum diseases among the three groups. Both the PG and SC cows had significantly lower culling rates than the HC cows (*p* = 0.012 and 0.041). The days in milk at the time of culling were 236 and 381 days for the NC cows, 200.6 ± 62.4 days (mean ± SEM) for the HC cows, 3 and 351 days for the PG cows, and 193, 263 and 496 days for the SC cows. No significant differences were observed in the incidence of clinical diseases within 60 DIM among the groups.

## 4. Discussion

This study investigated the effects of prophylactic administration of PG or SC to cows exhibiting elevated NEFA concentrations during the close-up period and compared their efficacy in improving NEB around parturition with that of the HC cows.

No significant differences in BCS were observed between any of the groups during the study period (Figure 1A). In contrast, the RFS was significantly higher in the PG cows than in the HC cows at 14 DIM (Figure 1B). Jeong et al. (2018) conducted a trial in which cows diagnosed with ketosis, defined as blood BHBA concentrations > 1.2 mM within four weeks after calving, were treated either with oral administration of PG alone or with a combination therapy consisting of oral PG, L-carnitine, and intravenous monensin for 3 to 5 days [35]. Their results demonstrated that RFS in the PG-containing treatment cows was significantly higher in the control cows at both 5 and 10 days after treatment. RFS reflects dry matter intake during the 12 h prior to measurement [36]. These results suggest that, in this trial as well, PG administration may have contributed to maintaining higher feed intake after parturition.

In the rumen, PG is metabolized to propionate, which is transported and absorbed by the liver, where it enters the TCA cycle and is ultimately converted to glucose through the gluconeogenesis pathway [16]. In addition, a portion of PG is metabolized primarily to lactate and pyruvate in the liver [37], and these metabolites are also utilized as gluconeogenic substrates via oxaloacetate. Insulin secretion is stimulated by increased blood glucose, which enhances lipoprotein lipase activity in peripheral tissues and promotes the uptake of blood TG into peripheral tissues [17]. Furthermore, insulin suppresses the activity of hormone-sensitive lipases in peripheral adipose tissues, thereby inhibiting the breakdown of stored fat [38]. Consequently, this leads to a decrease in the blood concentrations of NEFA [17]. From the outset, we anticipated that PG administration would lead to a rapid decrease in NEFA concentrations. However, blood NEFA concentrations increased in the PG cows, similar to those in the HC cows, until 3 DIM (Figure 1C). Fonseca et al. (2004) reported that daily administration of 300 mL of PG from 10 days prepartum until 16 days postpartum significantly decreased plasma NEFA concentrations 3 days postpartum [39]. Similarly, Lien et al. (2010) found that oral administration of 500 mL of PG mixed with 50 mL of molasses from 7 days prepartum to 30 days postpartum resulted in a sustained decrease in plasma NEFA concentrations from 3 days prepartum to 21 days postpartum [40]. The primary factor underlying the discrepancy between those previous findings and the present results may be differences in baseline NEFA concentrations at the start of treatment. In the cited studies, baseline NEFA concentrations ranged from 0.2 to 0.4 mEq/L [39,40], whereas the cows in the present study exhibited considerably higher levels (0.3–0.8 mEq/L), indicating more pronounced NEB. Therefore, in cows that already present with elevated NEFA concentrations as early as 14–7 days before parturition, as in the present study, short-term administration of PG may be inadequate to elicit a significant postpartum reduction in blood NEFA concentrations. Additionally, as shown in Table 2, the PG cows had the highest milk yield at the first DHI test after calving among the three groups, consistent with the findings of Stokes and Goff (2001), who reported increased milk yield following PG administration [41]. Such an increase in milk production may have exacerbated the NEB, leading to the elevated NEFA concentrations observed at 3 DIM. Furthermore, blood NEFA concentrations in the PG cows decreased from 7 DIM and were significantly lower than those in the HC cows by 14 DIM. Rukkwamsuk et al. (2005) reported that administering 400 mL/day of PG from 7 days prepartum to 7 days postpartum significantly reduced blood NEFA concentrations at 2 and 4 weeks postpartum [21]. Although the administration period in the present study was shorter than they reported, a certain reduction in NEFA concentrations during the early postpartum period was observed. Meanwhile, the higher RFS at 14 DIM in the PG cows compared to the HC cows is also considered to have contributed to the reduction in NEFA concentrations.

NEFA transported to the liver are converted into acetyl-CoA via β-oxidation. Acetyl-CoA then enters the TCA cycle, contributing to energy (ATP) production or gluconeogenesis. However, under conditions of pronounced NEB, a deficiency in carbohydrates results in reduced availability of oxaloacetate, which is essential for the proper functioning of the TCA cycle [16]. Consequently, excess acetyl-CoA that cannot be metabolized through the TCA cycle is redirected toward ketone body synthesis, leading to elevated concentrations of BHBA. Therefore, increased circulating NEFA concentrations are closely associated with enhanced BHBA production in dairy cows. Nielsen & Ingvartsen (2004) reported that cows with higher blood NEFA concentrations exhibited a more pronounced decrease in BHBA concentrations following PG administration [16]. However, in the present study, no significant differences in blood BHBA concentrations were observed between the PG and HC cows (Figure 1D). Grummer et al. (1994) orally administered 300–900 mL of PG over 5 days to primiparous cows subjected to 50% feed restriction [24], while Miyoshi et al. (2001) administered approximately 500 mL daily from 7 to 42 days postpartum [42]. Both studies involved cows with elevated NEFA levels and reported subsequent reductions in blood BHBA concentrations following PG treatment. However, these previous studies assessed BHBA changes only within 6 h post-administration. Furthermore, Lien et al. (2010) examined the effects of PG administration in high-NEFA cows, reporting that blood acetoacetate and BHBA concentrations were significantly lower in PG-treated cows than in control cows during the administration period [40]. PG administration in cows with high-NEFA concentrations has been confirmed to some extent on blood BHBA concentrations, but the duration of the effect may be limited in lactating cows. SC is rapidly hydrolyzed into monosaccharides by rumen microorganisms, leading to the production of volatile fatty acids (VFA) such as propionate and butyrate [18,43]. Therefore, like PG, SC is expected to help alleviate NEB by promoting gluconeogenesis and stimulating insulin secretion. In the present study, the SC cows showed a tendency toward lower blood NEFA concentrations at 3 DIM and lower BHBA concentrations at 21 DIM than the HC cows (Figure 1C,D). Although few previous studies have investigated SC feeding during the dry period, Ordway et al. (2002) reported that replacing ground corn with SC at 2.7% of dietary dry matter from 30 days before the expected calving date until calving did not affect postpartum NEFA concentrations [19]. Notably, changes in BHBA concentrations were not assessed in that study. The discrepancy between the results of the present study and their report regarding NEFA concentrations may be attributed to the markedly lower NEFA level (0.13 mEq/L) in the cows used in their study, which likely obscured the effect of SC administration. In contrast, the present study targeted cows with prepartum high-NEFA concentrations, and a suppression of the postpartum increase in NEFA concentrations was observed following short-term SC administration. Moreover, lower BHBA concentrations at 21 DIM indicated that SC administration contributed to alleviating the NEB.

At 3 DIM, blood glucose concentrations tended to be higher in the PG cows and were significantly higher in the SC cows than in the HC cows (Figure 1E). These findings suggest that even five-day administration of PG or SC to cows with elevated NEFA levels during the close-up period may promote the accumulation of gluconeogenic substrates in the liver, thereby helping to maintain blood glucose levels during the period of increased energy demand following calving. Additionally, blood insulin concentrations tended to be higher in the SC cows at 3 DIM and in the PG cows at 7 DIM than in the HC cows (Figure 1F). SC is rapidly and almost completely fermented in the rumen, with a reported hydrolysis rate of as high as 1.404% per hour [43]. Similarly, it is known that approximately half of the administered PG (100–910 g) disappears from the rumen within 1–2 h [44,45,46]. Mann et al. (2018) reported that increases in blood insulin and glucose concentrations following PG administration are generally mild and short-lived [47]. Previous studies monitoring blood biochemical parameters during the administration of both formulations from the prepartum period have demonstrated that insulin concentrations increase during the treatment period, but return to levels comparable to those of untreated controls once administration is discontinued [19,22,40]. In light of these findings, it appears that in the present study, both PG and SC were rapidly degraded in the rumen after administration and subsequently utilized by peripheral tissues, resulting in only limited direct stimulation of insulin secretion. However, the elevated insulin concentration observed in the treated cows in this study is presumed to reflect the enhanced insulin secretion stimulated by the higher blood glucose concentrations maintained postpartum as a result of these treatments.

Dairy cows experience a decline in insulin sensitivity during late pregnancy to prioritize glucose allocation to the fetus and mammary gland [48]. However, excessive reductions in insulin sensitivity have been reported to result in decreased postpartum productivity [49,50]. To evaluate insulin sensitivity in humans, the RQUICKI was developed [51,52], and the utility for assessing insulin sensitivity using it has also been reported in dairy cows [31]. A decrease in the RQUICKI value reflects a decline in insulin sensitivity and is considered to be an indicator of enhanced mobilization of adipose tissues [31]. Furthermore, excessive NEFA concentrations have been reported to reduce insulin sensitivity [53]. In the present study, the RQUICKI of the PG cows tended to be lower than that of the HC cows at 3 DIM (Figure 1G). However, the RQUICKI values in the early lactating stage for all cows, including changes showing tendencies, remained within the range of variation observed previously in healthy cows [31], and no important difference in insulin sensitivity due to the treatments was confirmed.

Serum AST activity was significantly lower in the SC cows than in the HC cows at 7 DIM (Figure 1H). Additionally, T-Bil concentrations were higher in the HC cows than in the PG and SC cows until 14 DIM (Figure 1I). Both AST and T-Bil are recognized as important indicators of hepatic function [54], and elevated values are commonly observed in cows with hepatic lipidosis compared to healthy individuals [55,56]. Vlizlo et al. (2021) reported that mean T-Bil concentrations at 2–3 weeks postpartum were approximately 0.2 mg/dL in healthy cows, whereas cows diagnosed with fatty liver exhibited significantly higher levels, averaging 1.0 mg/dL [57]. In the present study, the T-Bil concentrations in the HC cows reached 0.8 mg/dL at 3 DIM, a value approaching those reported in cases of hepatic lipidosis. Therefore, the suppression of T-Bil elevation observed in the PG and SC cows suggests that prophylactic administration of PG and SC may have the potential to alleviate postpartum hepatic lipidosis.

In addition to the pathway in which NEFA entering the liver are converted to acetyl-CoA via β-oxidation and subsequently enter the TCA cycle, fatty acids may also undergo re-esterification to form TG, which then associates with apolipoproteins and other lipid components to form VLDL. These VLDL are secreted into circulation and serve as an energy source for extrahepatic tissues [58,59]. VLDL is metabolized in the bloodstream by lipoprotein lipase (LPL) and converted to LDL via intermediate-density lipoproteins. LPL, which is located on the surface of vascular endothelial cells, functions to deliver the fatty acids contained in VLDL triglycerides to peripheral tissues. Furthermore, HDL plays a role in transporting excess cholesterol and phospholipids from peripheral tissues to the liver [60]. VLDL-protein and VLDL-total concentrations were higher in the PG cows than in the HC cows at 7 DIM (Figure 2D,E). Additionally, the LDL-TG concentration at 7 DIM was significantly higher in the PG cows than in the HC cows, with the SC cows also showing a tendency toward higher levels (Figure 2F). Oikawa et al. (1997) reported that in cows with fatty liver whose hepatic TG content exceeded a certain threshold, blood apolipoprotein B-100 (ApoB-100) concentrations decreased, which indicated a reduction in VLDL synthesis [61]. Additionally, Liu et al. (2014) treated cultured hepatocytes with various concentrations of NEFA and evaluated the effects on the synthesis and assembly of VLDL [62]. They found that the addition of NEFA significantly decreased the mRNA expression of ApoB-100 [62], the primary apolipoprotein constituting VLDL. Given that the elevated VLDL-protein concentration observed in the PG cows in this study likely reflects an increase in ApoB-100, it is therefore suggested that administration of PG or SC enhances the synthesis environment of VLDL. Furthermore, Studer et al. (1993) demonstrated that daily oral administration of 1L PG from approximately −10 ± 3.6 DIM significantly reduced hepatic TG content at both 1 DIM and 21 DIM [22]. Collectively, these findings suggest that PG administration may enhance hepatic VLDL secretion and thereby reduce hepatic TG accumulation. Uchida et al. (1992) demonstrated that administration of ethionine, a methionine analog, to non-pregnant cows markedly increased hepatic TG accumulation and significantly decreased ApoB-100 concentrations in both VLDL and LDL [60]. Furthermore, these decreases were partially correlated with reductions in TG concentrations within the respective lipoprotein fractions. Similarly, a decrease in the concentration of ApoA- I, the major apolipoprotein of HDL, and reductions in T-Cho and PL concentrations in both LDL and HDL were also observed. Notably, a decrease in the blood ApoA- I concentration has likewise been reported in cows with fatty liver [61]. These decreases in ApoB-100 and ApoA- I concentrations clearly reflected impaired lipid secretion from the liver, representing hepatic lipidosis. In the present study, the LDL-TG concentration in the HC cows at 7 DIM was significantly lower than that in the PG cows and tended to be lower than that in the SC cows (Figure 2F). Furthermore, HDL-TG and HDL-T-Cho concentrations at 3 DIM were higher in the PG cows than in the HC cows (Figure 2K,L). These findings suggest that administration of PG and SC may have the potential to mitigate hepatic lipidosis, with PG being potentially more effective, possibly through enhanced secretion of lipoproteins.

Enhanced oxidation of NEFA in the liver increases the production of reactive oxygen species (ROS), leading to elevated oxidative stress [12,63,64]. Oxidative stress results from an imbalance between lipid peroxides and antioxidant enzymes that neutralize them [65]. MDA, a byproduct of lipid peroxidation, is commonly used as a primary indicator of this process [66]. According to Senoh et al. (2019), cows that developed subclinical ketosis (SCK) had significantly higher serum MDA concentrations before and after calving than healthy control cows [13]. In the present study, MDA concentrations in all treated cows were higher than those in the NC cows at 14 DIM; however, no significant differences were detected among the three groups throughout the study period (Figure 3A). Although a reduction in the MDA concentration was anticipated with the preventive administration of PG or SC, no clear effect was observed. The average prepartum serum NEFA concentration in the cows used by Senoh et al. (2019) ranged from 0.22 to 0.24 mEq/L [13], whereas those in the cows included in the present study ranged from 0.3 to 0.8 mEq/L, indicating a more pronounced negative energy balance. Consequently, it is presumed that the extent of hepatic lipid peroxidation may have been greater in the cows of the present study.

PON-1, an antioxidant enzyme, is primarily synthesized and secreted by the liver [67] and circulates in serum bound to HDL [68]. PON-1 hydrolyzes lipids and thereby protects lipoproteins from oxidative stress [68,69,70]. Senoh et al. (2019) reported that serum PON-1 activity did not differ significantly between healthy control cows and SCK cows either before or after calving [13]. In contrast, Farid et al. (2013) reported that cows developing fatty liver after calving showed decreased PON-1 activity, as well as lower serum concentrations of LDL and HDL, indicating that PON-1 activity may be a useful marker for diagnosis of fatty liver [71]. In this study, PON-1 activity in the SC cows was lower than in the HC cows at 7 DIM, presumably due to its consumption in response to increased MDA concentrations in the SC cows (Figure 3B). Consequently, the lower AST activity and T-Bil concentrations, along with the higher LDL-TG concentrations observed in the SC cows at 7 DIM, are thought to reflect a reduction in oxidative stress in the liver.

In the analysis of milk production, the SC cows exhibited significantly higher MUN concentrations than the HC cows (Table 2). MUN is closely associated with ruminal protein metabolism in dairy cows and serves as an indicator of the energy-to-protein balance in the total dietary ration [72,73]. During periods of inadequate energy intake, the efficiency of converting ammonia produced from dietary protein in the rumen into microbial protein is reduced and consequently results in a decline in milk protein content. Additionally, excess ammonia that is not utilized is absorbed into the bloodstream and converted to urea, part of which transfers from the blood into the milk, resulting in elevated MUN levels [72]. Conversely, insufficient dietary protein intake reduces both microbial protein synthesis and milk protein content, simultaneously decreasing MUN concentrations. Therefore, the SC cows may have consumed more dietary protein over a prolonged period after calving than the HC cows.

Furthermore, a tendency for greater predicted 305-day milk yield was observed in the PG cows relative to the SC cows. Consistent with this observation, previous studies have shown that PG administered peripartum to NEB cows can increase average daily milk yield from calving to 90 DIM [40]. Given that prepartum administration of SC has been reported not to increase milk yield [19], these results suggest that preventive administration of PG to cows experiencing NEB may exert more prolonged effects, such as enhanced milk yield, than SC.

Ospina et al. (2010) have demonstrated that blood NEFA concentrations ≥ 0.29 mEq/L between 14 and 2 days before calving are associated with a 1.8-fold increased risk of developing periparturient disease [11]. The occurrence of such diseases has also been linked to an elevated risk of culling. Nicola et al. (2022) reported that cows with NEFA concentrations ≥ 0.26 mEq/L from 14 to 1 days prepartum had a 4.7-fold increased risk of being culled within 50 DIM [74]. Similarly, Chisato et al. (2024) found that cows with NEFA concentrations ≥ 0.31 mEq/L one week before calving had a 5.2-fold increased risk of culling within 60 DIM [9]. In the present study, PG and SC cows showed significantly reduced culling rates (Table 3). These results suggest that preventive administration of PG or SC during the close-up period to NEB cows may help reduce the risk of culling.

Additionally, prophylactic administration of SC, except for the 305-day predicted milk yield, showed effects comparable to PG in improving blood parameters and reducing culling rates. Therefore, considering its lower cost and easier accessibility, SC might it a more practical option for farmers whose primary goal is herd preservation rather than milk yield.

In the present study, we also attempted to investigate relations to reproductive performance. However, we could not obtain sufficient results including economic evaluation because the number of cases was limited (HC: *n* = 4, PG: *n* = 9, SC: *n* = 8) due to a lack of follow-up by culling. A limitation of this study is the relatively small sample size, which reduces the ability to draw firm conclusions, particularly for outcomes with smaller effect sizes such as periparturient disease incidences or reproductive performance. We regard this study as a preliminary trial and would like to conduct a larger-scale validation study to further confirm productivity performance, including the economic evaluation.

## 5. Conclusions

In conclusion, these results suggest that the preventive administration of PG or SC to cows with elevated NEFA concentrations before calving may reduce postpartum NEFA levels. Additionally, the SC cows showed a suppressive effect on the rise in BHBA concentrations. This intervention may be able to improve energy metabolism, potentially enhancing milk production, while also reducing the incidence of postpartum culling rates. Furthermore, the effects on blood metabolites and lipoprotein secretion were more pronounced in the PG administration, indicating a tendency toward greater efficacy compared with SC administration.

## Figures and Tables

**Figure 1 animals-15-03211-f001:**
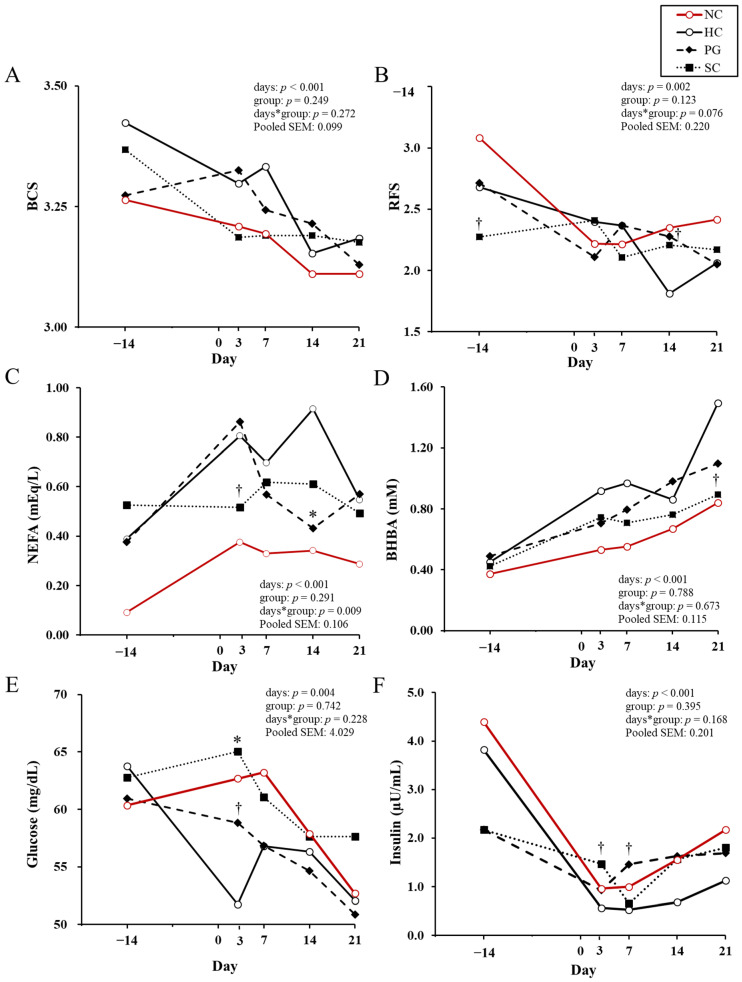
Physical monitoring scores and blood biochemical parameters before and after calving in each group. Least squares means are shown for the body condition score (BCS, (**A**)), rumen fill score (RFS, (**B**)), serum non-esterified fatty acids (NEFA, (**C**)), β-hydroxybutyrate (BHBA, (**D**)), blood glucose (**E**), insulin (**F**), revised quantitative insulin sensitivity check index (RQUICKI, (**G**)), aspartate aminotransferase (AST, (**H**)), and total bilirubin (T-Bil, (**I**)). NC group (○: negative control; healthy cows not included in the statistical analyses but shown as a reference; *n* = 15), HC group (○: positive control; untreated; *n* = 13), PG group (◆: 500 mL/day of propylene glycol administered; *n* = 11), SC group (■: 1000 mL/day of 50% sucrose solution administered; *n* = 11). * indicates a significant difference from the HC group on each sampling day (*p* < 0.05); † indicates a tendency toward significance (*p* < 0.10). A log_10_ transformation was applied to NEFA, BHBA and insulin for analysis. All obtained data were back-transformed.

**Figure 2 animals-15-03211-f002:**
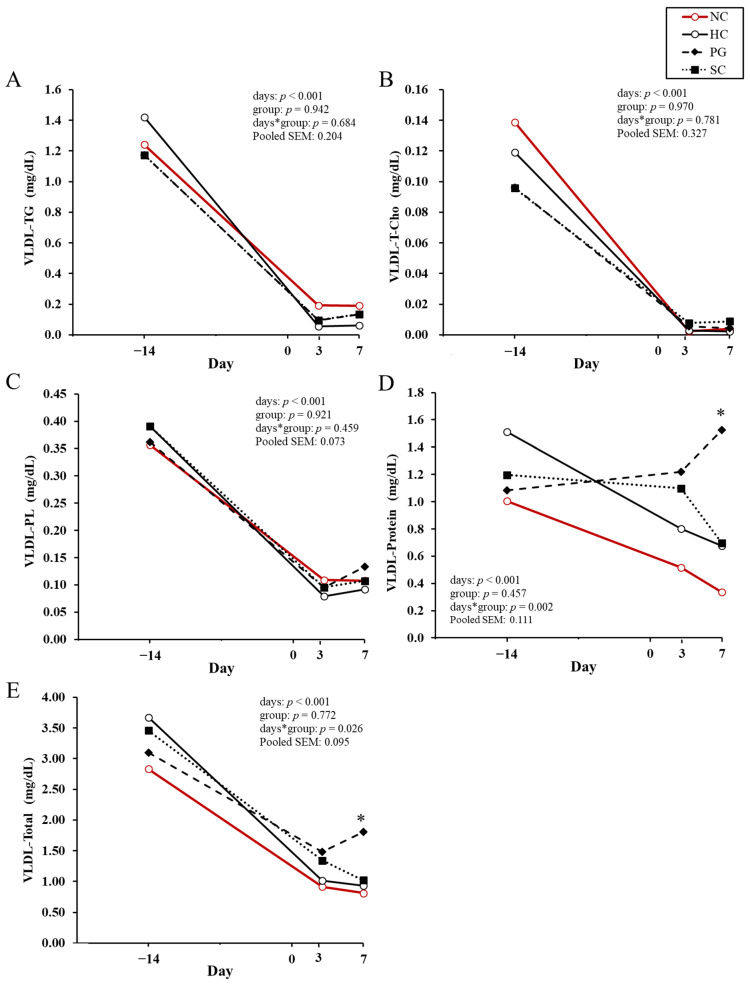
Lipoprotein components and total lipoprotein concentrations before and after calving in each group. Least squares means of triglycerides in VLDL (VLDL-TG, (**A**)), total cholesterol (VLDL-T-Cho, (**B**)), phospholipids (VLDL-PL, (**C**)), protein (VLDL-Protein, (**D**)) and total VLDL concentration (VLDL-Total, (**E**)); triglycerides in LDL (LDL-TG, (**F**)), total cholesterol (LDL-T-Cho, (**G**)), phospholipids (LDL-PL, (**H**)), protein (LDL-Protein, (**I**)) and total HDL concentration (LDL-Total, (**J**)); triglycerides in HDL (HDL-TG, (**K**)), total cholesterol (HDL-T-Cho, (**L**)), phospholipids (HDL-PL, (**M**)), protein (HDL-Protein, (**N**)) and total HDL concentration (HDL-Total, (**O**)) are shown. NC group (○: negative control; healthy cows not included in the statistical analyses but shown as a reference; *n* = 15), HC group (○: positive control; untreated; *n* = 13), PG group (◆: 500 mL/day of propylene glycol administered; *n* = 11), SC group (■: 1000 mL/day of 50% sucrose solution administered; *n* = 11). * indicates a significant difference from the HC group on each sampling day (*p* < 0.05); † indicates a tendency toward significance (*p* < 0.10). The total lipoprotein concentration represents the sum of TG, PL, T-Cho and protein values. Statistical analyses of each component concentration were conducted using log_10_-transformed data. All obtained data were back-transformed. ^a,b^ Different superscript letters indicate that groups show different trends over the sampling period (*p* < 0.10).

**Figure 3 animals-15-03211-f003:**
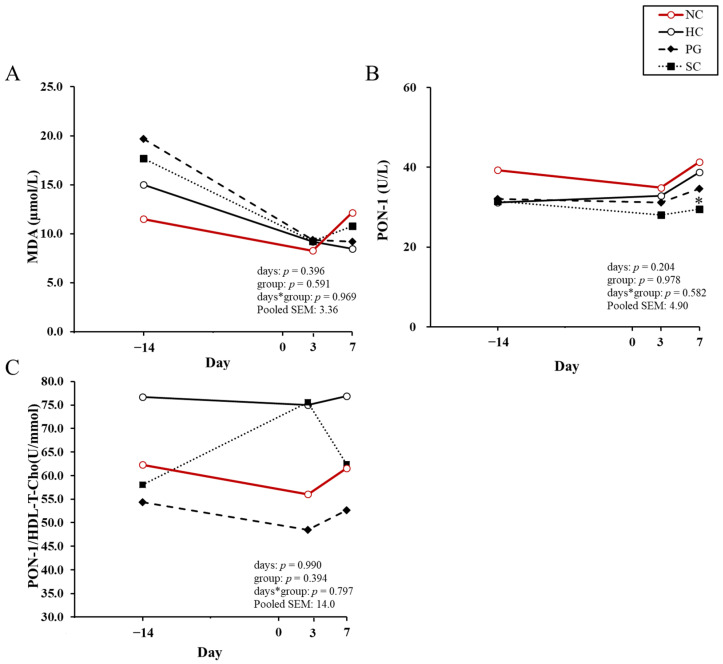
MDA, PON-1 and the ratio of total cholesterol in HDL to PON-1 before and after calving in each group. Least squares means of malondialdehyde (MDA, (**A**)), paraoxonase-1 (PON-1, (**B**)) and the ratio of total cholesterol in HDL to PON-1 (PON-1/HDL-T-Cho, (**C**)) are shown. NC group (○: negative control; healthy cows not included in the statistical analyses but shown as a reference; *n* = 15), HC group (○: positive control; untreated; *n* = 13), PG group (◆: 500 mL/day of propylene glycol administered; *n* = 11), SC group (■: 1000 mL/day of 50% sucrose solution administered; *n* = 11). * indicates a significant difference from the HC group on each sampling day (*p* < 0.05). In calculating PON-1/HDL-T-Cho, 0.02586 was multiplied by HDL-T-Cho as an adjustment factor [33].

**Table 1 animals-15-03211-t001:** Parity at the time of the initial blood sampling, actual days in milk, and duration of administration for each group at the time of the initial blood sampling.

Variable	NC ^1^	HC ^1^	PG ^1^	SC ^1^	*p*-Value
	*n* = 15	*n* = 13	*n* = 11	*n* = 11	
Parity ^2^	2.2 ± 0.3	2.8 ± 0.4	2.2 ± 0.5	2.7 ± 0.5	0.603
	(1.6–2.8)	(1.9–3.6)	(1.2–3.2)	(1.8–3.7)	
Actual days in milk at the time of initial blood sampling ^2^ (day)	−12.1 ± 0.7	−9.4 ± 0.9	−8.9 ± 1.0	−10.3 ± 1.0	0.636
	(−13.7–−10.6)	(−11.3–−7.5)	(−11.0–−6.8)	(−12.4–−8.2)	
Duration of administration ^2^ (day)	–	–	4.8 ± 0.1	4.7 ± 0.2	1.000

^1^ NC = negative control (healthy cows). This group was not included in the statistical analyses and is shown as a reference; HC = untreated control (positive control); PG = 500 mL/day of propylene glycol administered; SC = 1000 mL/day of 50% sucrose solution administered. ^2^ Data are expressed as mean ± SEM. Numbers in parentheses indicate 95% CI. Comparisons between groups were performed among HC, PG and SC using the Bonferroni multiple comparison test. *p*-values were calculated from comparisons among HC, PG and SC.

**Table 2 animals-15-03211-t002:** Milk yield and milk composition at the first postpartum DHI test for each group.

Variable	NC ^1^	HC ^1^	PG ^1^	SC ^1^	*p*-Value
	*n* = 15	*n* = 10	*n* = 10	*n* = 11	
Days in milk at the first DHI test ^2^ (day)	26.3 ± 4.5	27.2 ± 5.7	23.7 ± 5.7	35.3 ± 5.5	0.302
Milk yield ^2^ (kg)	36.2 ± 2.6	32.0 ± 5.0	38.4 ± 5.0	35.2 ± 4.9	0.432
Milk fat ^2^ (%)	4.3 ± 0.2	4.4 ± 0.2	4.4 ± 0.2	3.9 ± 0.2	0.214
Milk protein ^2^ (%)	3.3 ± 0.1	3.0 ± 0.1	3.2 ± 0.1	3.1 ± 0.1	0.303
De novo fatty acid ^2^ (g/100 g fat)	26.2 ± 0.9	22.6 ± 1.4	23.4 ± 1.4	26.4 ± 1.3	0.145
Preformed fatty acid ^2^ (g/100 g fat)	43.2 ± 1.7	49.0 ± 2.2	47.9 ± 2.2	42.6 ± 2.1	0.107
MUN ^2^ (mg/dL)	9.6 ± 0.4	7.8 ± 0.9 ^A^	9.5 ± 0.9 ^AB^	11.4 ± 0.9 ^B^	0.012
BHBA in milk ^2^ (mM)	0.03 ± 0.01	0.15 ± 0.04	0.07 ± 0.04	0.05 ± 0.03	0.123
305-day predicted milk yield ^2^ (kg)	12,310.1 ± 371.8	11,286.6 ± 518.6 ^ab^	12,110.0 ± 494.0 ^a^	10,531.0 ± 518.6 ^b^	0.067
4% fat corrected milk ^2,3^ (kg)	11,094.9 ± 278.9	10,620.3 ± 648.2	11,654.5 ± 642.0	10,105.6 ± 619.2	0.153

^1^ NC = negative control (healthy cows). This group was not included in the statistical analyses and is shown as a reference; HC = untreated control (positive control); PG = 500 mL/day of propylene glycol administered; SC = 1000 mL/day of 50% sucrose solution administered. Two cows in the HC group and one cow in the PG group were culled before the first postpartum DHI test and thus were excluded from the analysis. ^2^ Calculated from data at the first postpartum DHI test. Data are expressed as mean ± SEM. Comparisons between groups were performed among HC, PG and SC using the Bonferroni multiple comparison test. *p*-values were calculated from comparisons among HC, PG and SC. Statistical significance was considered at *p* < 0.05 for means with different uppercase letters, and a tendency was noted at *p* < 0.10 for means denoted by different lowercase letters. ^3^ The 4% fat-corrected milk (FCM) yield was calculated using the following formula [25]: FCM = (15 × milk fat percentage/100 + 0.4) × milk yield.

**Table 3 animals-15-03211-t003:** Culling rate and clinical disease incidence within 60 DIM for each group.

Variable	Group ^1^	% ^2^	OR	95%CI	*p*-Value
Culling rate ^3,4^	NC	13.3 (2/15)	–	–	–
	HC	69.2 (9/13)	Referent	–	–
	PG	22.2 (2/11)	10.1	1.47–69.94	0.012
	SC	27.7 (3/11)	6.0	1.02–35.38	0.041
Clinical disease incidence within 60 DIM ^3,5^	NC	30.0 (5/15)	–	–	–
	HC	61.5 (8/13)	Referent	–	–
	PG	27.2 (3/11)	4.3	0.75–24.18	0.093
	SC	54.5 (6/11)	1.3	0.26–6.81	0.527

^1^ NC = negative control (healthy cows). This group was not included in the statistical analyses and is shown as a reference; HC = untreated control (positive control); PG = 500 mL/day of propylene glycol administered; SC = 1000 mL/day of 50% sucrose solution administered. ^2^ The denominator indicates the number of animals evaluated, and the numerator indicates the number of affected animals. ^3^ The chi-square test was used to assess statistical significance, comparing each treatment group with the HC group. Differences were considered statistically significant at *p* < 0.05 and indicative of a trend at *p* < 0.10. ^4^ Culling included sold, slaughtered and died. The numbers of DIM at the time of culling in each group were as follows: 236 and 381 DIM in the NC group; 200.6 ± 62.4 days (mean ± SEM) in the HC group; 3 and 351 DIM in the PG group; and 193, 263, and 496 DIM in the SC group. ^5^ Including diseases occurring within 60 DIM.

## Data Availability

Data presented in this study are available upon request from the corresponding author.

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
