# Peer review of "Effects of Preventive Administration of Propylene Glycol or Sucrose in Dairy Cows with Elevated Blood Non-Esterified Fatty Acids During the Close-Up Period"

_animals, 2025, doi:10.3390/ani15213211_

Round 1

Reviewer 1 Report

Comments and Suggestions for Authors

General comments: This study aimed to evaluate the effect of dietary propylene glycol or sucrose (5 days prepartum) on dairy cows showing plasma levels of ≥0.3 mEq/L blood non-esterified fatty acids (NEFA) on energy/lipid metabolism and culling/disease rates during early postpartum. The novelty is related to the nature of samples (high plasma levels. Overall, this study is well presented and written. The major issue is related to the low sample size in each group (apparently low statistical power). Consequently, most of the results were presented as tendencies, including a few interactions (where the group is not significant). Since this is a non-modifiable factor regarding this specific experiment, I suggest clearly adding this limitation for an adequate interpretation, and to confirm this data or pattern in a further study with a higher sample size.  In fact, this issue represents a limitation in the scientific soundness. However, the results were clearly presented and supported by relevant tables and figures. Also, a great effort was made to compare the results with relevant literature and justify them. We can consider this study as preliminary results, which may indicate the direction of future research.

Specific comments

L17-19,35: Please see comments on L371

L27: 16-5=9.

L36-37: These results suggest that prophylactic administration of PG or SC in cows presenting NEFA plasma levels≥0.3 mEq/L during the close-up period improves energy metabolism … the PG group showed a more consistent effect.

L401-410: Overall, this is a repetition of L64-72.

In Fig. 2D and 2E, a group vs. time interaction is observed.

L371: It is not adequate to present these results as a tendency: note that the 95%CI varies from 0.75 to  24.18 (less than and more than 1). So, the likelihood cannot be determined (PG shows less or more chance to present postpartum diseases than the reference?).

L434-469: In Fig 1C: there is an interaction between group and time: PG and SG show different patterns for NEFA on days 3 and 14. Is there any additional justification?

L511-515: In Fig. 1I, the interaction between group and time is also significant. Can the potential changes in rumen microbiota due to PG and SC (less consistent effect) influence this pattern?

L525-526: In Fig. 2D and 2E, a group vs. time interaction is observed. Also here, the PG seems to have a more consistent effect.

L619-622: In this dietary protocol, it seems that PG was more advantageous.

Author Response

Thank you for your pertinent suggestions. Our revision is as the attached file.

Reviewer 2 Report

Comments and Suggestions for Authors

General Comments

This is a well-designed and executed study that addresses a question of high practical importance in the dairy industry.

The focus on cows with naturally occurring elevated NEFA levels in a field setting is a significant strength and a novel contribution to the literature.

The findings, particularly the significant reduction in postpartum culling rates following a simple 5-day intervention with either propylene glycol (PG) or sucrose (SC), are impactful and will be of great interest to veterinarians and dairy producers.

The manuscript is logically structured, and the methods are sound.

The following suggestions are intended to further strengthen the manuscript for publication.

Major Suggestions

Acknowledge Sample Size as a Limitation: The primary limitation of this study is the small sample size in each treatment group (n=11-13). While you were still able to detect significant effects for major outcomes like culling rate and some key blood metabolites, the low statistical power likely influenced other findings. For example, the reduction in postpartum disease incidence in the PG group was only a trend (). It is possible that with a larger cohort, this trend would have reached statistical significance. We strongly recommend adding a paragraph to the Discussion explicitly acknowledging the small sample size as a limitation, explaining that it restricts the ability to draw firm conclusions on outcomes with smaller effect sizes (like specific diseases or reproductive performance) and underscores the need for larger-scale validation studies in the future.

Minor Suggestions

Refine English Language and Phrasing: While the scientific meaning is clear, the manuscript would benefit from a thorough proofread to improve language flow and conciseness. Some sentences are slightly awkward or wordy.

Improve Visualization of Figure 2: Figure 2 presents a large amount of valuable data on lipoprotein components, but the format of 15 separate bar charts is visually dense and makes it difficult for the reader to discern trends over time. Consider reformatting this figure. One effective alternative would be to use line graphs (similar to Figure 1), perhaps grouped by lipoprotein type. For instance, you could have one multi-panel figure showing the changes in VLDL components (TG, T-Cho, PL, Protein) over time, with another for LDL and a third for HDL. This would greatly enhance the reader's ability to visualize and interpret the longitudinal changes in response to treatment.

Expand on Economic Implications in Discussion: You rightly note that SC is more readily available in Japan as it does not require a veterinary prescription. This is an interesting practical point. The discussion could be strengthened by briefly exploring the potential cost-benefit analysis for farmers. While PG-treated cows had a tendency for a higher predicted 305-day milk yield , both treatments were equally effective at reducing the culling rate. A short discussion on whether the lower cost and easier accessibility of SC might make it a more attractive option for farmers whose primary goal is herd preservation rather than milk yield maximization would add a valuable layer of practical insight.

Comments on the Quality of English Language

The scientific content of the manuscript is clearly conveyed.

However, in the opinion of this reviewer, the text would benefit from a thorough review by a native English speaker or a professional editing service. This would help to polish the prose by refining some awkward phrasing and improving overall sentence flow and concisenes, thereby strengthening the impact of this important research.

Author Response

Thank you for your pertinent comments. Our revision is as the attached file.

Round 2

Reviewer 1 Report

Comments and Suggestions for Authors

Dear Authors,

Thanks for providing this revised and improved version. The comments and suggestions of this reviewer were adequately addressed. Just in L36-37 (abstract) "... , with PG cows tending to have a lower incidence of postpartum disease..." can be removed. As discussed previously, the 95% CI was  0.75 – 24.18 (<1- >1) for P = 0.09. This issue was solved in the text, including in the conclusions. The mention of the low sample size as a limitation of this study is particularly welcome.